

# High field magnetism of the triangular lattice antiferromagnet CsFeCl$_3$ under high pressure

Katsuki Nihongi[1], Takanori Kida[1], Yasuo Narumi[1], Nobuyuki Kurita[2],
Hidekazu Tanaka[2], Yoshiya Uwatoko[3], Koichi Kindo[3] and Masayuki Hagiwara[1]⋆

**1** Center for Advanced High Magnetic Field Science (AHMF), Graduate School of Science, Osaka University, Toyonaka, Osaka 560-0043, Japan
**2** Department of Physics, Tokyo Institute of Technology, Meguro-ku, Tokyo 152-8551, Japan
**3** The Institute for Solid State Physics, The University of Tokyo, Kashiwa, Chiba 277-8581, Japan

⋆ hagiwara@ahmf.sci.osaka-u.ac.jp

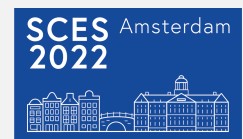

*International Conference on Strongly Correlated Electron Systems
(SCES 2022)
Amsterdam, 24-29 July 2022*

## Abstract

Magnetization measurements of the triangular lattice antiferromagnet CsFeCl$_3$ at 1.4 K for $H \parallel c$ were performed in pulsed high magnetic fields of up to 51 T under high pressures of up to 0.80 GPa. At ambient pressure, the magnetization below $H_{c1} = 4$ T is small, and then it changes steeply between $H_{c1}$ and $H_{c2} = 12$ T, followed by a gradual increase in magnetization up to $H_m = 33$ T where a two-step metamagnetic transition occurs. With increasing pressure, the $H_{c1}$ and $H_m$ shifted to the low-field side, while the $H_{c2}$ shifted to the opposite side. The change in $H_m$ may depend on the difference in the single-ion anisotropies on pressure in the lowest $J = 1$ and the excited $J = 2$ states.



## 1 Introduction

Gapped spin systems with a singlet ground state are attractive quantum spin systems, because the gap is suppressed by the applied magnetic field, pressure, or chemical doping, causing a quantum phase transition (QPT). In the vicinity of the quantum critical point (QCP) where the QPT occurs at zero temperature, some exotic phenomena are expected to be observed by the effect of quantum fluctuations.

CsFeCl$_3$ is one of the hexagonal ABX$_3$-type families that possesses the magnetic frustration in the triangular-lattice antiferromagnetic plane and belongs to a space group $P6_3/mmc$ [1]. Figure 1(a) shows the crystal structure of CsFeCl$_3$. Magnetic Fe$^{2+}$ ions, which are surrounded

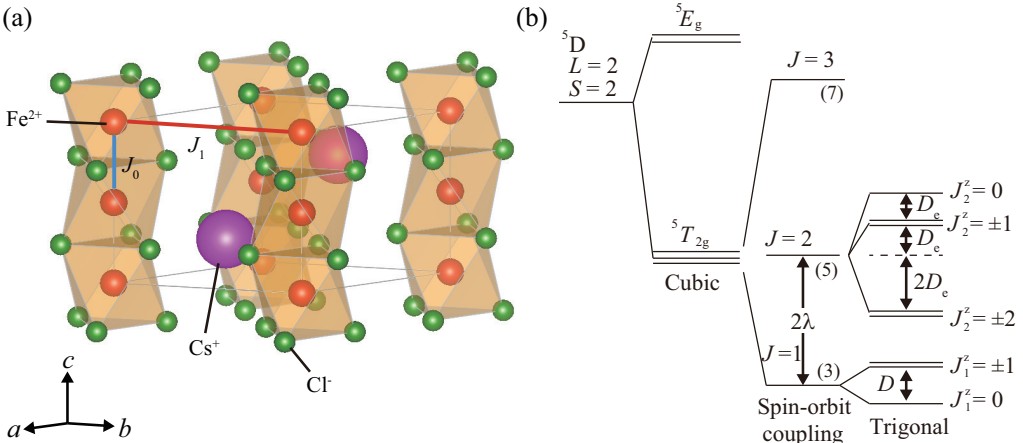

Figure 1: (a) Crystal structure of $CsFeCl_3$. Magnetic $Fe^{2+}$ ions are surrounded octahedrally by six $Cl^-$ ions. $FeCl_6$ octahedra form the chain along the $c$ axis by sharing Cl-triangular faces. The exchange interactions between $Fe^{2+}$ ions are denoted by the intra-chain ferromagnetic constant $J_0$ and the inter-chain antiferromagnetic constant $J_1$. (b) Expected energy scheme of $Fe^{2+}$ ion in $CsFeCl_3$.

octahedrally by six $Cl^-$ ions, form a chain structure with ferromagnetic (FM) exchange interactions ($J_0$) along the $c$ axis (chain direction) and a triangular lattice with antiferromagnetic (AFM) exchange interactions ($J_1$) in the $ab$ plane [2]. The energy scheme of $Fe^{2+}$ ion in $CsFeCl_3$ is depicted in Fig. 1(b). The $^5T_{2g}$ lowest state in a cubic crystalline field splits into three states owing to spin-orbit coupling. The energy gap between the lowest $J = 1$ state and the next excited $J = 2$ state is $2\lambda(\approx 200$ cm$^{-1})$ where $\lambda$ is the spin-orbit coupling constant of $Fe^{2+}$ ion. The $J = 1$ state consists of a singlet ground state and doublet excited state with a spin gap ($\Delta_s$) owing to easy-plane single-ion anisotropy. Considering only the energy scheme of the $J = 1$ state, the magnetic interaction terms of $Fe^{2+}$ ions (fictitious spin $S = 1$) in $CsFeCl_3$ in magnetic fields applied along the $z$ axis ($c$ direction) are described by the following Hamiltonian [3]

$$\mathcal{H} = \sum_i D(S_i^z)^2 + J_0 \sum_{<i,j>}^{\text{chain}} \boldsymbol{S}_i \cdot \boldsymbol{S}_j + J_1 \sum_{<i,k>}^{\text{plane}} \boldsymbol{S}_i \cdot \boldsymbol{S}_k - g\mu_B H \sum_i S_i^z, \tag{1}$$

where $D$ ($>0$) is the single-ion anisotropy constant, $J_0$ and $J_1$ are the ferromagnetic (along the chain) and antiferromagnetic (in the $ab$ plane) exchange constants, respectively. $g$ and $\mu_B$ in the forth term are the $g$ factor for the $z$ axis and the Bohr magneton, respectively. The coupling constants determined from the energy-dispersion relations in the inelastic neutron scattering (INS) experiments [4] are $D = 2.18$ meV, $J_0 = $ -0.454 meV, and $J_1 = 0.024$ meV. As aforementioned, $CsFeCl_3$ possesses a singlet ground state with a spin gap to the excited doublet state owing to this positive $D$.

With increasing applied pressure at zero field, $\Delta_s$ was reported to diminish and then vanish above $P_c = 0.90$ GPa to exhibit a magnetically ordered state with a 120° spin structure in the $ab$ plane at low temperatures [5, 6]. In the INS experiments of this compound under high pressure, the excitation modes, in which the longitudinal and transverse fluctuations of spins are hybridized, were observed at 1.4 GPa [7].

As for the magnetic-field experiments, the magnetization curve at 1.3 K for $H \parallel c$ increases gradually up to $H_{c1} = 4$ T, and then increases rapidly between $H_{c1}$ and $H_{c2} = 12$ T [8]. Neutron scattering experiments in magnetic fields revealed that the ground-state spin configuration is a 120° spin structure in the triangular-lattice plane between $H_{c1}$ and $H_{c2}$ [9]. Above $H_{c2}$,

the magnetization, increases slightly with magnetic field, which is attributed to the Van Vleck paramagnetism, and then exhibits a metamagnetic transition around $H_m = 33$ T [8]. This metamagnetic transition was suggested to be caused by an unconventional level crossing between the $J_1^z = -1$ and $J_2^z = -2$ states in magnetic fields as indicated in Fig. 1 (b). In this study, we performed high-field magnetization measurements of CsFeCl$_3$ at 1.4 K for $H \parallel c$ under pressures of up to 0.80 GPa and constructed the magnetic-field ($H$) versus pressure ($P$) phase diagram to discuss the pressure dependence of $H_{c1}$, $H_{c2}$ and $H_m$.

## 2   Experiment

Single crystals of CsFeCl$_3$ were grown by the vertical Bridgman method described in Ref. [5]. The sample was a cylindrical single crystal with  4 mm in length along the $c$ axis and  1.6 mm in diameter in the $ab$ plane. Pulsed high magnetic fields of up to 51 T with the duration of 35 ms were generated with a non-destructive pulsed magnet at AHMF in Osaka University. High-field magnetization measurements were performed with an induction method using a pick-up coil at the temperature of 1.4 K under high pressures of up to 0.80 GPa. In this study, we used a NiCrAl piston-cylinder-type pressure cell (PCC) with inner and outer diameters of 2.0 mm and 6.0 mm, respectively, and a mixture of Fluorinert FC70:FC77 = 1:1 as the pressure medium. The applied pressure in the sample space was calibrated by the pressure dependence of the superconducting transition temperature of Sn [10]. When putting the sample into the PCC, the sample could be adjusted by hand to apply the magnetic field to the $c$ axis of the sample within $\sim 4°$. The magnetization measurement of CsFeCl$_3$ at ambient pressure was performed without the NiCrAl PCC. Pulsed magnetic field induced eddy current in metallic parts of the NiCrAl PCC, resulting in the Joule heating. When the NiCrAl PCC is placed in liquid $^4$He cryostat at 1.4 K, the temperature at the sample position does not change much (less than 0.1 K) until 6.5 ms (approximately 40 T at this moment in the field ascending process when the maximum field was 51 T) from the start of magnetic-field generation [11], thus allowing high-field measurements up to $\sim$40 T at 1.4 K in the ascending process. It should be noted that the temperature of the pressure cell rises immediately, but the time delay of the heat flow via the pressure medium from the cell to the sample position occurs.

## 3   Results and Discussion

The magnetization ($M$) curves normalized by the $M$ at $H_{c2}$ and $dM/dH$ of CsFeCl$_3$ at 1.4 K for $H \parallel c$ under various pressures are shown in Figs. 2 (a) and (b), respectively. $M/M_{H_{c2}}$ and $dM/dH$ at ambient pressure are in good agreement with those in a previous paper [8]. The $M/M_{H_{c2}}$ around 33 T exhibits a metamagnetic transition with two steps at $H_{m1} = 32.2$ T and $H_{m2} = 32.9$ T, which corresponds to two peaks in $dM/dH$. With increasing pressure, the transitions fields $H_{c1}$, $H_{m1}$ and $H_{m2}$ shift to the low-field side, while the $H_{c2}$ shifts to the high-field side. The change in $M$ ($\Delta M$) or the slope between $H_{c2}$ and the magnetic field just below $H_{m1}$ was close to each other independent of pressure. Therefore, the Van Vleck paramagnetism is almost independent of pressure, suggesting weak pressure dependence of the spin-orbit coupling constant $\lambda$.

Figure 3 shows the magnetic field versus pressure phase diagram of CsFeCl$_3$ at 1.4 K for $H \parallel c$. The calculated formula for $H_{c1}$ within the mean-field theory [12] is given by,

$$H_{c1} = \Delta_s / g\mu_B = \sqrt{D^2 - 2D(2J_0 + 3J_1)}/g\mu_B. \tag{2}$$

The pressure dependences of the exchange interaction and the single-ion anisotropy deter-

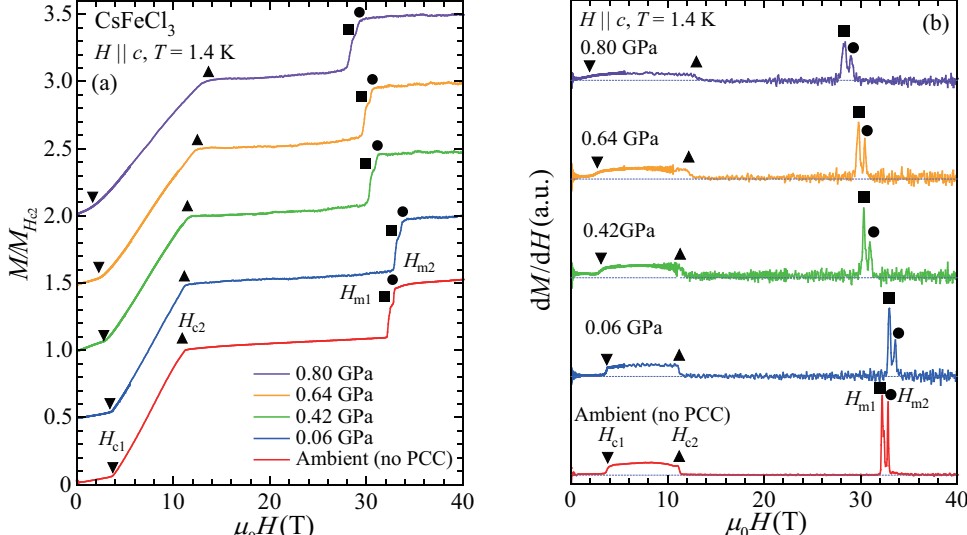

Figure 2: (a) Magnetization ($M$) curves of CsFeCl$_3$ at 1.4 K for $H \parallel c$ normalized by the $M$ at $H_{c2}$ under various pressures. The curves are shifted up by 0.5 from the ambient-pressure curve with increasing pressure for clarity. (b) The field derivatives of the magnetization ($dM/dH$) curves in (a). The $dM/dH$ curves in (b) are shifted arbitrarily along the vertical direction for clarity.

mined from the INS study [7] were described as follows; $J_0$ (meV) = -0.5 -0.14$P$, $J_1$ (meV) = 0.0312 -0.0015$P$, and $D$ (meV) = 2.345 +0.365$P$, where the $P$ (GPa) is the value of pressure. The dashed line in Fig. 3 expresses the pressure dependence of $H_{c1}$ given by Eq. (2), in which the pressure-dependent exchange constants and the single ion anisotropy constant given above are substituted. The comparison of the experimental data to the calculated one is made and these are closely placed to each other. The $H_{c1}$ from our magnetization measurement, however, is slightly higher than that obtained from the INS experiment at high pressures. This might be attributed to the quantum fluctuation enhanced near the QCP ($P_c$=0.9 GPa), because Eq. (2) is derived under the mean field. The spin states of Fe$^{2+}$ ion with the fictitious spin $S = 1$ in CsFeCl$_3$ are composed of a singlet-ground state ($S^z = 0$) and the doublet-excited state ($S^z = \pm 1$) separated by a single-ion anisotropy $D$ as shown in Fig. 1(b). However, the excited state has a finite energy band that becomes broad with increasing pressure. Between $H_{c1}$ and $H_{c2}$, the lower doublet state with a finite bandwidth comes down to cross the singlet ground state. Therefore, the $H_{c1}$ goes down and the $H_{c2}$ goes up with increasing pressure, indicating the increases in the single-ion anisotropy and the bandwidth of the energy dispersion curve, the former of which is consistent with the pressure dependence of the single-ion anisotropy as mentioned above.

The transition fields $H_{m1}$ and $H_{m2}$ decrease almost linearly with increasing pressure as shown in Fig. 3. These extremely low transition fields could not be explained by the level crossing of $J = 1$ and $J = 2$ states in a conventional energy scheme of Fe$^{2+}$ ion in Fig. 1(b), and its origin is unknown yet. The pressure dependence of the metamagnetic transition field may be explained by the pressure effect on the single-ion anisotropy $D$ and the single-ion anisotropy $D_e$ in the excited $J = 2$ quintet states as shown in Fig. 1(b). These $D$ and $D_e$ values may change largely by $\delta/\lambda$ where $\delta$ is the parameter that describes the splitting of the orbital triplet state $d\epsilon$ by the tetragonal crystalline field as discussed for FeCl$_2$·2H$_2$O in Fig. 2 of Ref. [3]. To clarify this interpretation, the pressure dependence of the energy difference between $J_1^z = \pm 1$ in $J = 1$ state and $J_2^z = \pm 2$ in $J = 2$ needs to be investigated by INS and ESR measurements.

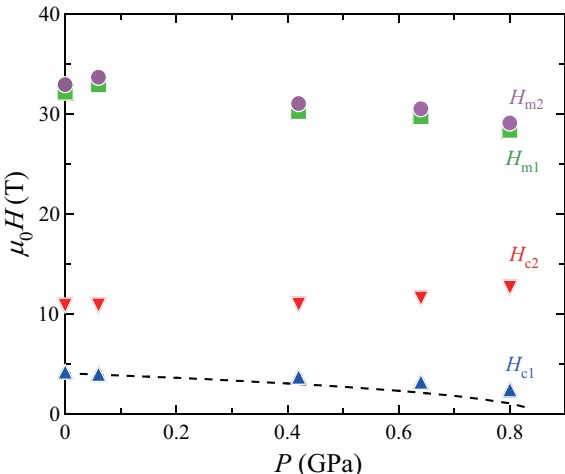

Figure 3: Magnetic field ($\mu_0 H$) vs pressure ($P$) phase diagram of $CsFeCl_3$ at 1.4 K for $H \parallel c$ constructed from the high-field magnetization measurements. A dashed line is the pressure dependence of $H_{c1}$ provided in a previous study [5].

## 4  Summary

We performed magnetization measurements of $CsFeCl_3$ at 1.4 K in pulsed high magnetic fields under high pressures. The pressure dependence of $H_{c1}$ is explained fairly by the formula Eq. 2 using the pressure-dependent values of the exchange constants and single-ion anisotropy constant obtained from the INS experiment under pressures. The metamagnetic transition field around 33 T decreases linearly with pressure, which may be explained by the change in the single-ion anisotropy $D$ in the $J = 1$ state and the single-ion anisotropy $D_e$ in the excited $J = 2$ state under pressure.

## Acknowledgements

This study was supported by the Sasakawa Scientific Research Grant from The Japan Science Society, JST, the establishment of university fellowships towards the creation of science technology innovation, Grant Number JPMJFS2125 and the Motizuki Fund of Yukawa Memorial Foundation, and also supported by JSPS KAKENHI (Grant Nos JP17H06137, JP17K18758, 19K03711, JP21H01035, JP19H00648).

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
