# Peer review of "High field magnetism of the triangular lattice antiferromagnet CsFeCl3 under high pressure"

_SciPost Physics Proceedings, doi:SciPost Phys. Proc. 11, 012 (2023)_

## Round 1 · Referee Report · Anonymous · 2023-1-6

Strengths
- Well written.
- Nice results.
Report
This article meets the criteria for the SCES conference proceedings.
Requested changes
1: In Fig. 2 it is not clear how the magnetization curves are normalized. It appears that curves at finite pressure are offset from zero pressure, but this is not clearly stated.
2: It is stated that the temperature of the cell does not change much. Can the authors be more quantitative?
3: This is a layered material. The accuracy of aligning the field along the c-axis is stated as to be 4 degrees. Can the authors comment on how this influences the results? In particular, can this explain splitting of the meta-magnetic transition and deviation of the measured Hc1 compared to Eq. 2? I’m thinking of decomposing the field in in-plane and out-of-plane components. This would lead to larger discrepancies at larger fields.
4: The Hc1 line determined from INS data deviates from the measured Hc1 at larger fields. Can the authors comment on (or add an additional curve) how the agreement between Eq. 2 and then new data can be improved? Which parameter(s) in Eq. 2 can improve the agreement?
Author: Katsuki Nihongi on 2023-01-17 [id 3242]
(in reply to Report 1 on 2023-01-06)Dear Referee,
We thank you very much for reading our manuscript (scipost_202207_00035v1) carefully and giving us useful and instructive comments. We have responded to your comments, as described below.
1: In Fig. 2 it is not clear how the magnetization curves are normalized. It appears that curves at finite pressures are offset from zero pressure, but this is not clearly stated.
Thank you for pointing out this matter. In the caption of Fig2(a), we add the sentence “The curves are shifted up by 0.5 from the ambient-pressure curve with increasing pressure for clarity.”
2: It is stated that the temperature of the cell does not change much. Can the authors be more quantitative?
We change the corresponding sentence to “When the NiCrAl PCC is placed in liquid ^4He cryostat at 1.4 K, the temperature at the sample position does not change much (less than 0.1 K) until 6.5 msec (approximately 40 T at this moment in the field ascending process when the maximum field was 51 T) from the start of magnetic field generation [11], thus allowing high-field measurements up to ~ 40 T at 1.4 K in the ascending process. It should be noted that the temperature of the pressure cell rises immediately, but the time delay of the heat flow via the pressure medium from the cell to the sample position occurs.
3: This is a layered material. The accuracy of aligning the field along the c-axis is stated as to be 4 degrees. Can the authors comment on how this influences the results? In particular, can this explain splitting of the meta-magnetic transition and deviation of the measured Hc1 compared to Eq. 2? I’m thinking of decomposing the field in in-plane and out-of-plane components. This would lead to larger discrepancies at larger fields.
We found the peaks at H_m1 and H_m2 were merged into one peak when the direction of the magnetic field deviated more than four degrees from the c axis. Therefore, in this study, the field direction should deviate less than four degrees from the c axis. With this deviation, the influence on the single ion anisotropy seems to be quite small, because the DS_z^2 changes with cos2θ(θ=0 corresponds to the c (z) axis and less than 0.5 % atθ=4 degrees), resulting in a tiny shift of H_c1.
4: The Hc1 line determined from INS data deviates from the measured Hc1 at larger fields (pressures?). Can the authors comment on (or add an additional curve) how the agreement between Eq. 2 and then new data can be improved? Which parameter(s) in Eq. 2 can improve the agreement?
The dashed line in Fig. 3 was drawn with Eq. 2 derived within the mean field theory using the pressure-dependent parameters obtained from the INS experiment. Accordingly, this equation is not applicable at high pressure near P_c, because the quantum fluctuation, which is not considered in the mean field theory, is enhanced. We add some sentences in the text to describe this matter. According to the referee’s request, we checked the fitting by changing the parameters in Eq. (2). As a result, if we change the coefficient of P in the parameter D from 0.365 to 0.4 (about 10% larger), the agreement between experiment and calculation becomes better as shown Supplementary Figure 1. But this change without reasonable explanation is thought to be meaningless.
We modified some sentences and words in the text. In addition, the D_e in Fig. 1(b) has changed because of the definition of D_e (D_eS_2^z2 : S_2^z is actually J_2^z). We believe that we respond to all the comments by the referee correctly and our paper is worth being published in JPS Conference Proceedings.
Sincerely Yours, Masayuki Hagiwara
AHMF (Center for Advanced High Magnetic Field Science), Graduate School of Science, Osaka University, Machikaneyama 1-1, Toyonaka, Osaka 560-0043, Japan
Attachment:

---

## Round 2 · Author Response

Dear Referee,

We thank you very much for reading our manuscript (scipost_202207_00035v1) carefully and giving us useful and instructive comments.

We submit re-upload manuscript.

Sincerely Yours,

Masayuki Hagiwara

---

## Round 2 · List of Changes

1. In the caption of Fig2(a), we add the sentence “The curves are shifted up by 0.5 from the ambient-pressure curve with increasing pressure for clarity.”

2. The D_e in Fig. 1(b) has changed because of the definition of D_e (D_eS_2^z2 : S_2^z is actually J_2^z)

---

## Editorial Decision

published